# Core–Shell Spheroid Structure TiO_2_/CdS Composites with Enhanced Photocathodic Protection Performance

**DOI:** 10.3390/ma16113927

**Published:** 2023-05-24

**Authors:** Tingting Chen, Bo Li, Xiaolong Zhang, Xiang Ke, Rengui Xiao

**Affiliations:** 1School of Chemistry and Chemical Engineering, Guizhou University, Guiyang 550025, China; chentingting3642@163.com (T.C.); zhangxl628@163.com (X.Z.); kexiang91@163.com (X.K.); 2Institute of Electric Power Science of Guizhou Power Grid Co., Guiyang 550001, China; gzgylb2207@163.com

**Keywords:** core–shell structure, spheroid, composite coating, photocathodic protection

## Abstract

In order to improve the conversion and transmission efficiency of the photoelectron, core–shell spheroid structure titanium dioxide/cadmium sulfide (TiO_2_/CdS) composites were synthesized as epoxy-based coating fillers using a simple hydrothermal method. The electrochemical performance of photocathodic protection for the epoxy-based composite coating was analyzed by coating it on the Q235 carbon steel surface. The results show that the epoxy-based composite coating possesses a significant photoelectrochemical property with a photocurrent density of 0.0421 A/cm^2^ and corrosion potential of −0.724 V. Importantly, the modified composite coating can extend absorption in the visible region and effectively separate photoelectron hole pairs to improve the photoelectrochemical performance synergistically, because CdS can be regarded as a sensitizer to be introduced into TiO_2_ to form a heterojunction system. The mechanism of photocathodic protection is attributed to the potential energy difference between Fermi energy and excitation level, which leads to the system obtaining higher electric field strength at the heterostructure interface, thus driving electrons directly into the surface of Q235 carbon steel (Q235 CS). Moreover, the photocathodic protection mechanism of the epoxy-based composite coating for Q235 CS is investigated in this paper.

## 1. Introduction

With rapid development in human society, metal materials have emerged as one of the most extensively utilized engineering materials in modern society. Despite their widespread usage, metal corrosion has posed a significant challenge for humans. The detrimental effects of metal corrosion are manifold, including substantial economic losses, resource and energy waste, and environmental harm. Therefore, taking some effective measures to prevent or suppress metal corrosion has great social value and important research significance [1]. Among the various metal anti-corrosion technologies, cathodic protection stands out as one of the most effective methods. As a new type of electrochemical metal anti-corrosion technology, photocathodic protection was proposed by Tsujikawa and co-workers in 1995 [2]. It has attracted wide attention from researchers due to its low cost, environmental friendliness, simple technology, and other advantages. The principle of photogenerated cathodic protection rests on the excitation of electrons in the valence band (VB) of the photoelectric material, which are transitioned to the conduction band (CB) upon the absorption of solar energy and transferred to the metal surface, modulating the potential of the photoelectric material below the metal self-corrosion potential and finally realizing the anti-corrosion function of the metal [3,4]. Although researchers have made many new compound materials (such as ZnO [5], NiSe_2_/TiO_2_ [6], PbS/TiO_2_ [7], ZnPc/TiO_2_ [8], and Ag/SnO_2_/TiO_2_ [9]) aimed at improving the photoelectric conversion performance in photocathodic protection, it is always one of the important research directions.

In recent years, semiconductor materials (such as Fe_2_O_3_ [10], ZnO [11], CuBi_2_O_4_ [12,13], and Sb_2_Se_3_) have been found to exhibit a favorable photoelectric response [14]. Apart from these materials, TiO_2_ is often used as a photoanode material in photocathodic anti-corrosion strategy due to its excellent chemical stability, light resistance, non-toxicity, durability, and low cost. However, TiO_2_ solely absorbs ultraviolet light, has a wide band gap (3.2 eV), and photogenerated holes and electrons are easily recombined in the dark state, which inhibits its broad applications. Therefore, TiO_2_ needs to be modified to improve the separation efficiency of photogenerated electron–hole pairs and broaden the photoresponse range. At present, TiO_2_ modification methods include doping by metal or non-metal ions [15,16], coupling with semiconductors [17,18,19,20,21,22] or nano-carbon materials [23,24,25,26,27], polymer modification [28,29], and so on. To extend the photo-response of TiO_2_ to the visible-light region, considerable efforts have been taken to couple TiO_2_ with semiconductors. Among the various inorganic semiconductors, CdS as an n-type semiconductor is more promising due to its low cost and appropriate band edge position for the TiO_2_ photocathodic protection reaction.

CdS is a semiconductor characterized by a narrow band gap width of 2.4 eV. It not only exhibits excellent optical absorption performance in the visible light region [30,31,32] but also has well-matched energy levels with TiO_2_, which can improve the injection efficiency of photogenerated electrons. Therefore, a composite’s heterojunction system is fabricated through assembled CdS, a sensitizer, with pure TiO_2_ to improve the visible light absorption of TiO_2_ and excite more electron–hole pair separation, thus sufficiently inhibiting the recombination of photogenerated carriers and effectively ameliorating its photoelectric performance [33,34,35]. Unlike previous studies on TiO_2_/CdS composites, this paper employs TiO_2_-coated CdS to prepare the composite coating. This TiO_2_/CdS composite possesses a core–shell structure with a tight interface of the CdS core and TiO_2_ shell layer. The TiO_2_ particles in the outer layer can inhibit the precipitation of Cd^2+^ ions and protect CdS from corrosion by light or other reagents to achieve excellent corrosion resistance and environmental performance. Meanwhile, because of the unique core–shell spheroid structure, TiO_2_/CdS composites can improve the charge transfer efficiency due to the driving force of the internal electrostatics, thus effectively ameliorating their photoelectric performance. Although materials have been commonly used as anti-corrosive fillers [36,37,38], the core–shell spheroid structure TiO_2_/CdS composites differ from a layered-structure anti-corrosion filler [39,40]. In addition, the TiO_2_/CdS composite material is considered to be the filler of epoxy-resin-based coatings to enhance the performance of photocathodic protection, which can effectively optimize the effect of metal anti-corrosion properties of metals. The strategy proposed in this paper will provide a beneficial exploration route for metal anti-corrosion.

## 2. Materials and Methods

### 2.1. Preparation of CdS Nanomaterials

Firstly, 0.01 mol CdCl_2_ was added to 0.25% Cetyltrimethylammoniumbromide (CTAB) aqueous solution (20 mL) and was ultrasonicated for 15 min. Next, 0.01 mol Na_2_S was also added to another 0.25% CTAB aqueous solution (20 mL) and ultrasonicated for 15 min. Then, the solution containing Na_2_S was mixed with another solution containing CdCl_2_ and it was stirred in a closed environment for 2 h. The mixed solution was transferred to a hydrothermal reactor and reacted for 12 h at 120 °C. Afterward, the CdS nanomaterials were filtered, washed with deionized water/ethanol, and then dried at 70 °C for 6 h.

### 2.2. Preparation of TiO_2_/CdS Composites

Figure 1 is the schematic diagram of composite synthesis. Briefly, 0.01 mol CdS was added to 0.25% CTAB ethanol solution (20 mL) and stirred for 30 min. Subsequently, a little deionized water was added and it continued to be stirred for 30 min. In addition, 0.01 mol of tetrabutyl stitanate was added to 20 mL ethanol in a closed environment. After stirring for 30 min, it was poured into a beaker with CdS and stirred for 2 h. The composite coatings were named TiO_2_/CdS−1, TiO_2_/CdS−2, TiO_2_/CdS−3, TiO_2_/CdS−4, and TiO_2_/CdS−5, corresponding to the mass ratios of tetrabutyl titanate to cadmium sulfide, which were 1:1, 2:1, 5:1, 7:1, and 10:1, respectively. Finally, all mixed solutions were transferred to a hydrothermal reactor and reacted for 12 h at 120 °C, and the TiO_2_/CdS composite materials were filtered, washed with deionized water/ethanol, and then dried at 80 °C for 6 h.

### 2.3. Preparation and Application of TiO_2_/CdS-Modified Epoxy Resin Composite Coatings

A steel plate with a thickness of 1 mm, a length of 15 mm, and a width of 10 mm was used. The natural oxide film on the steel plate was removed with 250 mesh silicon carbide paper and then degreased with acetone before use. The samples were rinsed with distilled water, degreased with acetone, dried at room temperature, and stored in a vacuum desiccator. Under normal conditions, 0.1 g TiO_2_/CdS composites were added to ethanol and dispersed via ultrasound for 20 min to form a suspension of the composite. In this way, the composites can be easily dispersed and aggregated. The suspension was added into 10 g epoxy resin (E44), stirred at high speed for 1 h, and then placed in an oven at 70 °C to remove the moisture from the composites. Then, an appropriate amount of 1-methyl-2-pyrrolidone (NMP) was added to improve the dispersibility of the materials. The temperature was kept at 50 °C and it was stirred thoroughly for 1 h to completely disperse the materials in the epoxy resin. The mass ratio of curing agent (T31) and material epoxy resin was mixed 1:3, and then it was evenly brushed onto the surface of Q235 CS and cured naturally at room temperature (25 ± 2 °C) for 48 h. The thickness of the coating was 80 ± 3 μm.

### 2.4. Characterizations

The morphology of the TiO_2_/CdS composites was observed using a field emission scanning electron microscope (SEM, Hitachi Regulus 8100, Tokyo, Japan) and transmission electron microscope (TEM, JEM 1400 Flash, Tokyo, Japan). The phase composition of the TiO_2_/CdS composite materials was analyzed using an X-ray diffractometer (XRD, X’PertProMPD, Almelo, The Netherlands). A UV-2700i ultraviolet-visible diffuse reflectance spectrometer (UV−Vis DRS, Shimadzu, Kyoto, Japan) was used to measure the light absorption characteristics of the composite materials with a scanning range of 200–800 nm (BaSO_4_ as reference). The functional group of the composite was analyzed using a Nicolet-iS5 Fourier Transform Infrared Spectrometer (FT−IR, Thermo Fisher Scientific, Waltham, MA, USA). The photoelectron–hole separation capabilities of the prepared powder were assessed using a photoluminescence spectrometer (PL, Varian Cary Eclipse, Palo Alto, CA, USA).

### 2.5. Photoelectric Performance Test

The photoelectrochemical properties of the Q235 CS coupled with the composite coating were characterized using the electrochemical workstation (CHI760E, Chenhua, Shanghai, China). As shown in Figure 2, a typical three-electrode system was used for electrochemical analysis. The saturated calomel (SCE) was used as the reference electrode, Pt was used as the counter electrode, Q235 CS coupled with composite coating was used as the working electrode, and 0.2 mol/L sodium hydroxide (NaOH) and 0.1 mol/L sodium sulfide (Na_2_S) were used as the electrolytes. The light source was a 300 W xenon lamp (PLS−SXE300UV, Beijing, China) and AM1.5 was simulated sunlight. At 0 V (Hg/HgO) bias potential, the photocurrent density curve and the open-circuit potential (OCP) curve were tested. In addition, electrochemical impedance spectroscopy (EIS) was carried out with a frequency range from 100 kHz to 0.01 Hz, the amplitude was 5 mV, and a scanning rate of 10 mV/s was used. The Tafel curves were also measured at a potential scan rate of 0.5 mV/s.

## 3. Results

### 3.1. Microstructure Characterization

SEM and TEM were performed to observe the microstructure of the CdS and TiO_2_/CdS composites as well as to explore the combining form between CdS and TiO_2_. Pure CdS exhibited sphere-like particles (Figure 3a), pure TiO_2_ exhibited a high density of fine particles (Figure 3b), and TiO_2_/CdS showed many spherical and irregular TiO_2_ particles being loaded onto the surface of the CdS particles (Figure 3c). After the introduction of TiO_2_, the TiO_2_ particles were wrapped around the surface of the CdS sphere-like particles, forming a thin layer, which implied the successful construction of the core–shell structure (Figure 3d,e). HRTEM images showed 0.35 nm and 0.335 nm crystalline surface spacing, which was attributed to the (101) crystalline surface of TiO_2_ and the (002) crystalline surface of CdS (Figure 3f). The EDS images of TiO_2_/CdS (Figure 3g) indicate that TiO_2_/CdS consists of elements Cd, S, Ti, and O. This further confirms the formation of thin TiO_2_ layers on the surface of CdS. To investigate the TiO_2_/CdS heterojunction, elemental mapping was performed, showing that TiO_2_ was successfully immobilized on the CdS surface, resulting in a tight interface between the CdS core and the TiO_2_ shell (Figure 3h). As shown in Figure 3h, the diameters of the Ti and O element distributions were significantly larger than those of the Cd and S element distributions, further confirming the construction of the core–shell structure. The TiO_2_/CdS sample formed a core–shell structure with a tight interface, which was beneficial for improving the separation and transfer efficiency of the photoinduced electron–hole pairs. Importantly, the outer TiO_2_ particles can inhibit the precipitation of Cd^2+^ ions and protect CdS from corrosion by light or other reagents. Thus, the environmental protection effect and photocatalytic activity of the TiO_2_/CdS composites can be improved.

### 3.2. Physicochemical Characterization of TiO_2_/CdS Composites

The crystalline structure of the TiO_2_/CdS composites was analyzed via X-ray diffraction (XRD). Figure 4a shows the crystal structures of the TiO_2_/CdS−1, TiO_2_/CdS−2, TiO_2_/CdS−3, TiO_2_/CdS−4, and TiO_2_/CdS−5 samples. The sharp and intense diffraction peaks indicate that the TiO_2_/CdS samples are well crystallized. Among them, the diffraction peaks appeared at 2θ values of 25.28°, 36.95°, 48.01°, 53.89°, 55.06°, and 68.72°, corresponding to (101), (103), (200), (105), (211), and (116) crystal planes of TiO_2_ (JCPDSNO.21−1272), respectively [41]. In addition, there were also obvious diffraction peaks at 24.8°, 26.3°, 28.2°, 36.46°, 43.8°, 47.9°, and 52°, which were indexed as (100), (002), (101), (102), (110), (103), and (112) crystal planes of CdS (JCPDS NO. 75−1545), respectively [42]. The above results confirmed that all of the TiO_2_/CdS composites were prepared successfully. Additionally, the peaks of TiO_2_ were not shifted when CdS was introduced into TiO_2_, indicating that the crystal structure of TiO_2_ was barely changed by the heterogeneous process.

In addition, the TiO_2_, CdS, and TiO_2_/CdS samples with different TiO_2_ contents were tested and analyzed using Fourier infrared spectroscopy (FT−IR). As shown in Figure 4b, each sample had a wide absorption peak at ~3440 cm^−1^, which corresponds to the stretching vibration peak of the hydroxyl group (−OH). The flat head peak that appeared at 500−800 cm^−1^ was the stretching vibration and bending vibration of the Ti−O−Ti bond, which is the absorption characteristic peak of TiO_2_. In addition, the absorption peaks at 1630 cm^−1^ and 1120 cm^−1^ were attributed to the stretching vibration absorption peak of Cd−S. At the same time, all of the TiO_2_/CdS samples had characteristic absorption peaks of CdS and TiO_2_ at 1630 cm^−1^, 1120 cm^−1^, 620 cm^−1^, and 515 cm^−1^. This result fully confirmed that CdS and TiO_2_ form a composite system. The results were consistent with the above-mentioned XRD test.

Moreover, the light absorption properties of TiO_2_/CdS samples with different TiO_2_ contents were characterized using UV−Vis DRS. As shown in Figure 4c, the TiO_2_ sample had a strong absorption band in the wavelength range of 350~400 nm, and the absorption edge was around 390 nm, which represented the typical feature of TiO_2_. However, compared with the absorption edge of TiO_2_, the absorption edge of the TiO_2_/CdS composite sample had a significant shift to the visible light region. It is well known that CdS has a certain absorption capacity in both ultraviolet and visible regions, and has a strong absorption band in the wavelength range of 550~600 nm [43]. In addition, because of the strong interaction between electrons within the TiO_2_/CdS heterojunction composite, the separation efficiency of the photoelectron−hole pairs was enhanced and the spectral absorption range was extended; thus, a new optical absorption edge appeared. However, when the loading content of TiO_2_ was too much, it would limit the absorption of visible light by CdS. In contrast, the formation of the TiO_2_/CdS heterojunction composite was less, decreasing the binding capacity to single electron oxygen holes, thus affecting the range of visible light absorption. Thus, the cut-off light absorption edge of the TiO_2_/CdS−3 sample was the largest (~574 nm) in this study. Moreover, the bandgap of the as-prepared sample could be calculated according to the Kubelka Munk function [44,45]. As shown in Figure 4d, the gapband energy values of the TiO_2_, CdS, TiO_2_/CdS−1, TiO_2_/CdS−2, TiO_2_/CdS−3, TiO_2_/CdS−4, and TiO_2_/CdS−5 samples were 3.22, 2.1, 2.83, 2.92, 2.16, 3.01, and 2.57 eV, respectively. Among them, the gapband energy of the TiO_2_/CdS−3 sample was the smallest, which indicated that it possessed excellent visible light absorption capacity.

The PL spectroscopy test was an effective technique to characterize the charge carrier trapping and transfer on the semiconductor materials’ surface. As shown in Figure 5, the highest peak position of the fluorescence emission of the TiO_2_/CdS samples was around 425 nm and intensity gradually decreased due to the introduction of CdS to promote photoelectron−hole separation. The intensity of the PL emission peaks of the TiO_2_/CdS samples was much weaker than that of the pure TiO_2_ and CdS; this indicated that the formation of a TiO_2_/CdS heterojunction composite could further impede the recombination of photogenerated charge carriers due to the presence of the built-in electric field between CdS and TiO_2_ at the heterogeneous interface. In addition, the TiO_2_/CdS−3 sample had the lowest fluorescence intensity, which suggested the highest electron–hole pair separation efficiency. The higher the electron−hole pairs’ separation efficiency, the more electrons were injected into the metal surface, indicating that the coating was more effective in protecting the metal.

### 3.3. Photoelectrochemical Performance

Figure 6a shows the change curve of photocurrent density with an irradiation time of different TiO_2_/CdS composite coatings with intermittent visible light being on and off. The higher the photocurrent density was, the more photogenerated electrons were generated by the composite under the action of light, and the more electrons were transferred to the surface of Q235 CS, thus achieving better photocathodic protection [46]. Among them, CdS photo-corrosion results in the formation of low-surface-activity products that occupy the active surface of the coating to absorb daylight, and thus the photoelectrochemical reaction may be reduced [47]. However, it is known from Figure 6a that the composite coatings all have excellent photocurrents. This is because TiO_2_ covers CdS and inhibits the photo-corrosion of CdS. The photocurrent densities of all of the samples immediately changed once the light was turned on or off, indicating the high sensitivity of the composite coating to the light. More importantly, TiO_2_/CdS had good periodicity and repeatability after multiple switching lamp experiments, and the photocurrent response did not change, which indicated that the composite coatings possessed excellent stability. However, the photocurrent density produced by TiO_2_/CdS−3 was the highest (0.0421 A/cm^2^). Moreover, the photocurrent density of TiO_2_/CdS−4 (0.0296 A/cm^2^) and TiO_2_/CdS−2 (0.0188 A/cm^2^) was lower than that of TiO_2_/CdS−3 (0.0421 A/cm^2^), indicating that more TiO_2_ or less TiO_2_ loading will affect the visible light absorption performance, thus resulting in a decline in photoelectric performance. This result could be attributed to the core−shell spheroid structure of the TiO_2_/CdS heterojunction composite. The structure had a special electron transfer path, which could sufficiently inhibit the recombination of photogenerated carriers. This was consistent with the PL spectroscopy results, where the lowest fluorescence intensity indicates the highest efficiency of electron–hole pair separation. It means that the greater the possibility of the composite coating providing electrons to the metal surface under light conditions, the better it will be for metal anti-corrosion. Therefore, the TiO_2_/CdS−3 possessed superior photocathodic protection for Q235 CS.

Figure 6b shows that the actual anti-corrosion property of the composite coating for Q235 CS could be evaluated using the OCP−time curve. Under the irradiation of xenon lamp conditions (simulating solar light), photoelectrons and holes were separated, resulting in a circuit which generated an opposite voltage to the original space charge layer potential. Therefore, the more negative the OCP of the prepared coatings, the better the photocathodic protection performance of the photoelectric electrode. As shown in Figure 6b, the TiO_2_/CdS has quick photoresponse conduct. The dotted line is the self-corrosion potential of Q235 CS (−0.58 V). Additionally, the corrosion potential of the composite coatings would decrease rapidly first, and then slowly become stable after lighting. However, when the light was turned off, the potential of the composite coatings could slowly rise again, indicating that the recombination between electron and hole was a slow process [11,48,49,50,51,52]. In addition, under light illumination, the corrosion potential of TiO_2_/CdS−3 decreased rapidly from −0.621 V (vs. SCE) to −0.724 V (vs. SCE), and the photovoltage change value reached 103 mV. Moreover, the change values of the photovoltage of TiO_2_/CdS−4 (65 mV) and TiO_2_/CdS−2 (50 mV) were lower than those of TiO_2_/CdS−3. This indicates that lower or higher TiO_2_ loading will affect the visible light absorption performance. So, the photocathodic protection effect of the composite coatings would decline. The corrosion potential of TiO_2_/CdS−3 was negatively shifted to −0.724 V (vs. SCE) after illumination conditions, indicating that the TiO_2_/CdS−3 composite coating provides better photoelectric cathodic protection for Q235 CS. Compared with the photoelectrodes in Table 1, the CdS significantly improved the photoelectric performance of bare TiO_2_, and the TiO_2_/CdS composite coating had excellent photoelectric performance. These results were consistent with the PL test results.

Figure 7 shows the negative migration of the corrosion potential of Q235 CS because the TiO_2_/CdS composite can provide electrons to the Q235 CS surface and inhibit Q235 CS corrosion, since a large number of photoelectrons were injected into the surface of the Q235 CS, causing the polarization potential of the metal to drop sharply when the light source was turned on. This result indicated that the self-corrosion potential of the composite coating moved negatively and the self-corrosion current density increased, thus realizing the photocathodic protection of composite coatings for Q235 CS. According to the analysis, the self-corrosion potential of the composite coatings had a negative shift before and after illumination, which was attributed to the TiO_2_/CdS core−shell heterojunction system being able to effectively separate photogenerated carriers. These experimental results were in agreement with the SEM and TEM test results. More importantly, the TiO_2_/CdS−3 sample had the largest negative shift change and most negative corrosion potential as well as the largest corrosion current density in the Tafel polarization curve under the light illumination. This result demonstrated that the TiO_2_/CdS−3 sample had the best photocathodic protection effect for Q235 CS under illumination conditions. These results were also consistent with those of the OCP test.

Figure 8 presents typical electrochemical impedance spectroscopy (EIS) plots and fits of equivalent circuits for bare Q235 CS and Q235 CS coupled with different TiO_2_/CdS composite coatings under light illumination. In the proposed equivalent circuit model (Figure 8d), Rs, Rcoat, Qcoat, Rct, and Cdl represent electrolyte resistance, coating resistance, coating capacitor, charge transfer resistance, and electric double-layer capacitor, respectively. Under the irradiation of a xenon lamp simulating sunlight, photoelectrons and holes were separated, while the arc diameters of the Nyquist plots represent the electron transfer resistance between the working electrode and the electrolyte interface. It is well known that the smaller diameters of the arc, the higher the transfer efficiency between the electron and hole pair [53]. However, the arc diameters of all of the test samples were as follows: Q235 > TiO_2_/CdS−1 > TiO_2_/CdS−2 > TiO_2_/CdS−5 > TiO_2_/CdS−4 > TiO_2_/CdS−3 (Figure 8a). Under illumination conditions, the arc diameters of the TiO_2_/CdS composite coatings were smaller than those of bare Q235 CS. This was due to the injection of photogenerated electrons from the sample into Q235 under illumination, which accelerates the electrochemical reaction at the interface of the sample and Q235, and shows the positive effect of the TiO_2_/CdS composite in improving charge separation and suppressing electron–hole pair complexation. The smaller arc radius, which shows its higher charge transfer efficiency, indicates that the sample has better protection performance against Q235 under visible light. Among them, the impedance of the TiO_2_/CdS−3 sample was the smallest, which indicates that TiO_2_ and CdS completely combine to form a unique core–shell heterostructure in the TiO_2_/CdS−3 system. More importantly, the heterojunction system could bind a large number of single electron oxygen vacancies, and there was a strong interaction between the electrons, which improved the separation and transfer efficiency between the photogenerated electrons and hole pairs. In the phase diagram (Figure 8b) and Bode diagram (Figure 8c), the samples with different additions exhibit a shift in the frequency region of the capacitive behavior, and the log|Z|_10mHz_ values decrease for all samples, indicating enhanced electron transfer efficiency, a faster transfer to the surface of the Q235 CS substrate, and increased photogenerated cathodic protection performance of metals.

The fitting results of Rs, Cdl, and Rct are displayed in Table 2. As shown in Table 2, the Rct values of all of the TiO_2_/CdS composites are much smaller than those of Q235, indicating that all of the TiO_2_/CdS composites have excellent charge transfer efficiency. The results are in good agreement with those of OCP and photogenerated current density. The photogenerated electrons are transferred from the CB of CdS to the CB of TiO_2_. Then, the electrons are transferred to the surface of Q235 CS, resulting in a lower OCP of the steel. In other words, the heterojunction between TiO_2_ and CdS contributes to the separation and transport of photogenerated electrons and holes. In addition, TiO_2_/CdS−3 has the smallest Rct value, indicating that TiO_2_/CdS−3 exhibits the most effective separation of photogenerated carriers.

### 3.4. Anti-Corrosive Properties of Coatings

Tafel polarization curves enable the corrosion protection of coatings against metals to be evaluated. The self-corrosion current density (I_corr_), polarization resistance (Rp), self-corrosion voltage (E_corr_), corrosion protection efficiency (PEF%), and corrosion rate (CR) can be used to characterize anti-corrosion performance. The polarization resistance Rp can be calculated using the Stern–Geary equation [54] and the Tafel linear extrapolation method, as in Equation (1).
(1)$$Rp=\frac{k_{a}\cdot k_{b}}{2.303\left(k_{a}+k_{b}\right)I_{corr}}$$
where I_corr_ is the self-corrosion current density (A/cm^2^), and k_a_ and k_b_ are the Tafel slope of cathode and anode, respectively (ΔE/Δlogi); anti-corrosion coatings generally have a low corrosion rate (CR) [55]. CR is calculated as
(2)$$CR=k\times \frac{M_{m}I_{corr}(A/{cm}^{2})}{{n\rho }_{m}(g/{cm}^{3})}$$
where M is the molecular weight, I_corr_ is the self-corrosion current density, K is the constant 3270, n is the number of electrons lost in the oxidation process, and ρ is the density; PEF% [56] can be calculated by Equation (2):(3)$$PE\%=\frac{R_{\left(uncoated\right)}^{-1}-R_{\left(coated\right)}^{-1}}{R_{\left(uncoated\right)}^{-1}}\times 100\%$$

Figure 9 showed the electrochemical Tafel polarization curve of bare Q235 CS and Q235 CS coupled with different TiO_2_/CdS composite coatings under darkness. In addition, the corrosion potential of the composite coatings was slightly shifted to the right and the corrosion current density was reduced under the dark conditions, which indicated that the anti-corrosion performance of the epoxy resin coating for Q235 CS was improved after the TiO_2_/CdS composite materials were modified. Moreover, the barrier performance of the epoxy resin coating was enhanced when the TiO_2_/CdS composites were used as fillers to modify it. The specific surface area of the composites was increased after TiO_2_ coating with CdS. TiO_2_/CdS composites do not have the same specific surface area as graphene oxide, but they also provide a zigzag diffusion path and form a physical protective barrier layer for Q235 CS [57].

The corrosion parameters calculated from the Tafel plots for the TiO_2_/CdS anti-corrosive coatings are summarized in Table 3. The CR of the TiO_2_/CdS−3 coated CRS was found to be 2.669 × 10^−3^ mm year^−1^, which was 4.87 times lower than that observed for neat CRS (1.3 × 10^−2^ mm year^−1^). Among them, the corrosion prevention efficiency of the TiO_2_/CdS composite coating reached 85.51%.

From Figure 10a, it is observed that the surface of the pure epoxy coating has more micropores, which is because the epoxy resin produces many holes during the curing process, and the corrosive medium can simply corrode the metal substrate surface through the coarse layer with low anti-corrosion performance. As observed from Figure 10b, compared with the pure epoxy coating, there are no obvious holes on the surface of the TiO_2_/CdS composite coating, indicating that the surface of the TiO_2_/CdS composite coating is more dense, which is because the TiO_2_/CdS composite material fills some holes of the epoxy coating as a filler, making the corrosion path more tortuous and increasing the difficulty of the corrosive medium corroding Q235 CS.

The immersion test was a simple and effective strategy for visualizing the coating anti-corrosion process. The coating with defects of about 5 cm was exposed to the mixture solution of electrolytes. The relevant pictures were taken at different time intervals. As shown in Figure 11, there was obvious rust appearing in the scratch area for the pure epoxy resin coating group after 10 days (Figure 11a). This phenomenon could be attributed to the metal oxidation reaction caused by the corrosion medium penetrating into the Q235 CS substrate. With the exposure time increasing, the color of the surrounding rust was darker, and the rust layer almost filled the area around the scratch. The TiO_2_/CdS composite coating group with or without light illumination also showed a similar dynamic trend to the pure epoxy resin group (Figure 11a,b). However, in the dark conditions, the TiO_2_/CdS composite coating group showed only slight corrosion products after 10 days, indicating that the holes of the epoxy resin were filled by TiO_2_/CdS composite during the solvent volatilization. In addition, the epoxy-resin-based composite coating could form a physical barrier to prevent the propagation of electrolytes and delay the corrosion of Q235 CS by the corrosion medium [58]. Importantly, the TiO_2_/CdS-modified composite coating only produced a small amount of corrosion products after 10 days with light illumination (Figure 11c), and the corrosion products were less than those of it under dark conditions and the pure epoxy resin group after 15 days. This shows that the electron and hole pairs in the TiO_2_/CdS heterostructures were separated under light illumination. Moreover, the photoelectrons generated by the conduction band (CB) of TiO_2_ and CdS would provide electrons for the surface of Q235 CS, thus achieving a photocathodic protection function for Q235 CS [59]. These observation results were very consistent with the EIS test under light illumination.

Toxic leaching properties are of great importance for industrial solid wastes, especially coatings. Here, we investigated the release of cadmium (Cd) ions from TiO_2_/CdS composite coatings after mixing in the brine solution. The TiO_2_/CdS composite coating released 0.15 ppm of Cd^2+^ ions after 48 h of immersion. The Cd^2+^ ion release was lower than the limit value (1 ppm) specified by GB 5085.3-2007 [60]. The results indicate that Cd is well encapsulated in the TiO_2_/CdS core–shell structure. Therefore, the use of TiO_2_ as the shell and CdS as the core to prepare the composite coating of the TiO_2_/CdS core–shell structure is an environmentally friendly choice.

## 4. Discussion

Figure 12 showed the proposed photocathodic protection mechanism of TiO_2_/CdS composite heterostructures for Q235 CS. The epoxy resin coating could isolate the invasion of corrosive medium in metal protection and protect the metal from the impact of an external corrosive environment [61]. However, during the curing process, microporosity is created inside the epoxy coating due to solvent evaporation, leading to the corrosion medium experiencing immersion from the surface into the interior of the Q235 CS and finally inducing corrosion. In contrast, when the epoxy resin was modified by TiO_2_/CdS composites, the holes formed during the curing process could be filled to improve the compactness of the as-prepared coating. This epoxy-resin-based composite coating would form a dense protective barrier for carbon steel and finally prevent the corrosion of Q235 CS. The wide band gap of TiO_2_ leads to the absorption of UV light only and high carrier compounding efficiency; so, CdS-sensitized TiO_2_ was introduced. When a TiO_2_/CdS composite heterostructure is produced, the light absorption range is expanded, thus improving the separation efficiency between photogenerated carrier pairs. As shown in Figure 12b,c, the electron and hole pairs in the TiO_2_/CdS heterostructures would separate when the composite coating was exposed to sunlight. Interestingly, TiO_2_ could form a tight contact interface because of the special core–shell spheroid structure, leading to better contact with water, which increases the active sites on the surface of the composite material and promotes uniform dispersion. Meanwhile, because of the potential energy difference between the Fermi energy and the excitation energy level, the photoelectrons generated by the CB of TiO_2_ and CdS would quickly transfer to the metal surface and this would occur during the reduction reaction, making the corrosion potential negatively shift, while the holes would form during the oxidizing reaction. This means that the TiO_2_/CdS composite coating can provide electrons and plays a key role in photocathodic protection for Q235 CS.

## 5. Conclusions

In this paper, a core–shell heterostructure TiO_2_/CdS composite was synthesized using the simple hydrothermal method in situ. The epoxy resin was modified by the TiO_2_/CdS composite and used as a composite coating to realize the photocathodic protection function for Q235 CS. In addition, the physicochemical properties of the composite were analyzed via FT−IR, XRD, UV−Vis DRS, and PL. The results showed that the photoelectrochemical properties of the epoxy-resin-based composite coating were improved significantly due to the broadening of the light absorption range and the improvement in the photoelectron–hole pair separation efficiency. In addition, a physical shielding effect was formed on the Q235 CS surface under the synergistic effect between the TiO_2_/CdS composite and the epoxy resin, which significantly improved the corrosion resistance for Q235 CS. After being modified by the TiO_2_/CdS composite, the epoxy-resin-based composite coating possessed higher photocurrent density (0.0421 A/cm^2^) and negative corrosion potential (−0.724 V). More importantly, the fabricated binary heterojunction composite coating had excellent stability and good photoelectrochemical properties. The design route proposed in this paper has great potential to be applied in the field of anti-corrosion engineering.

## Figures and Tables

**Figure 1 materials-16-03927-f001:**
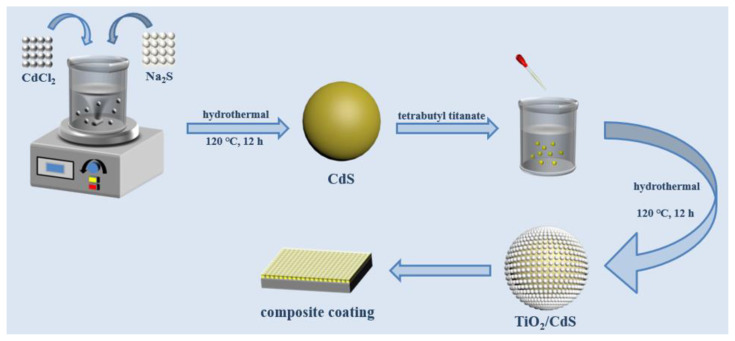
Schematic diagram of TiO_2_/CdS composite fabrication.

**Figure 2 materials-16-03927-f002:**
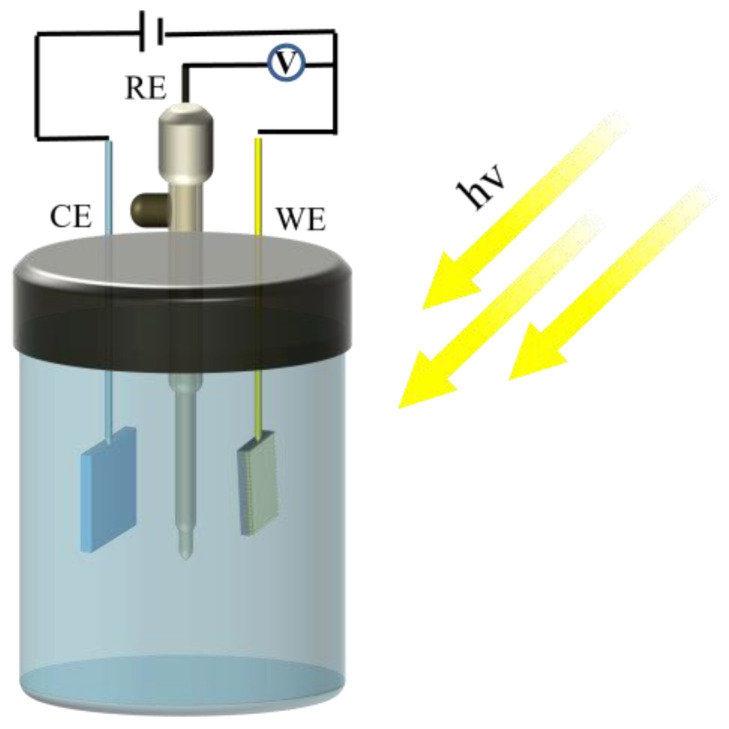
Schematic diagram of testing device for photoelectric performance.

**Figure 3 materials-16-03927-f003:**
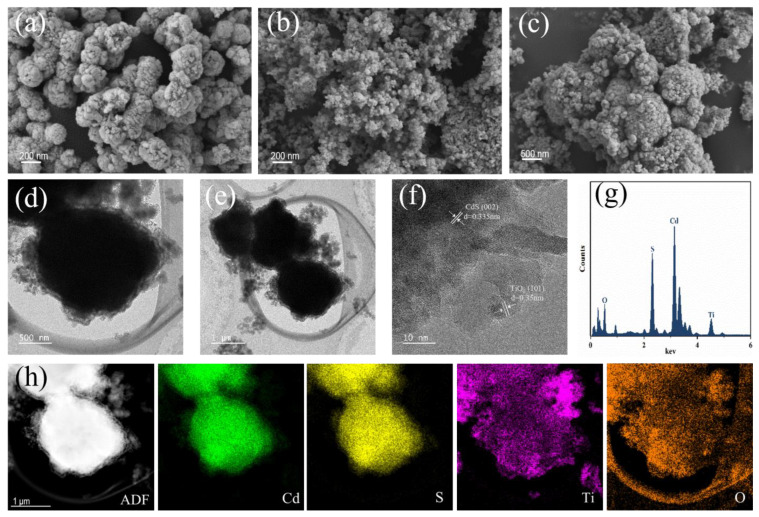
The SEM images of CdS, TiO_2,_ and TiO_2_/CdS, respectively (**a**–**c**); the TEM images (**d**,**e**), HRTEM image (**f**), and EDS (**g**); and the elemental mapping (**h**) of TiO_2_/CdS.

**Figure 4 materials-16-03927-f004:**
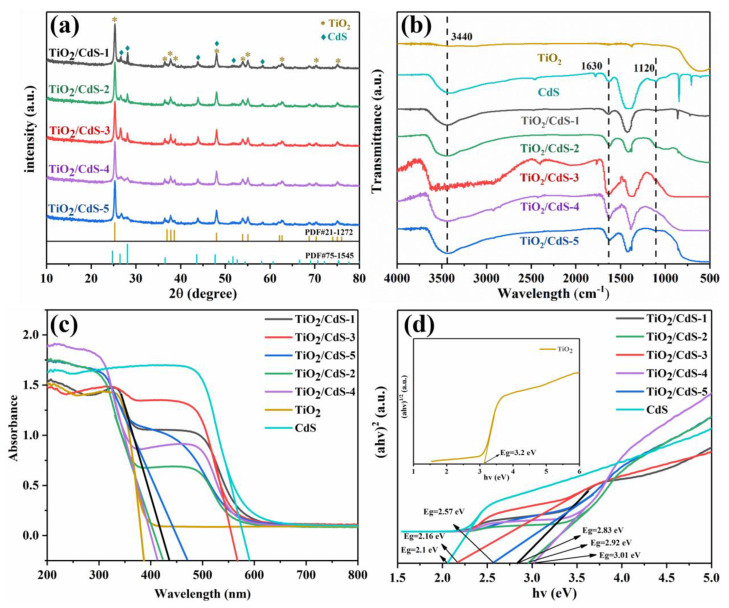
(**a**) XRD pattern of TiO_2_/CdS composite; (**b**) FT−IR spectra of TiO_2_, CdS, and TiO_2_/CdS composites with different TiO_2_ contents; (**c**,**d**) are UV−vis diffuse reflectance spectra and the corresponding bandgap of TiO_2_ and TiO_2_/CdS composites with different TiO_2_ contents, respectively.

**Figure 5 materials-16-03927-f005:**
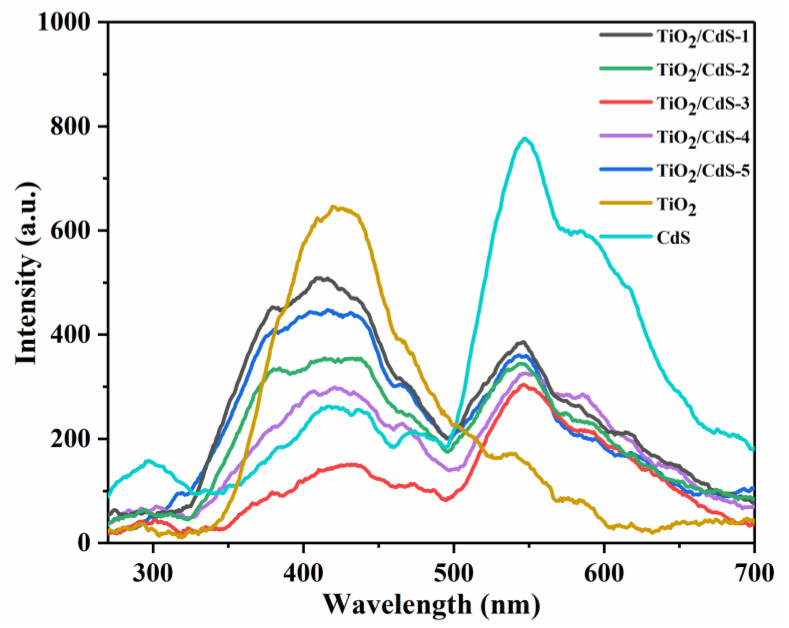
PL spectra of TiO_2_, CdS, and TiO_2_/CdS with different TiO_2_ contents.

**Figure 6 materials-16-03927-f006:**
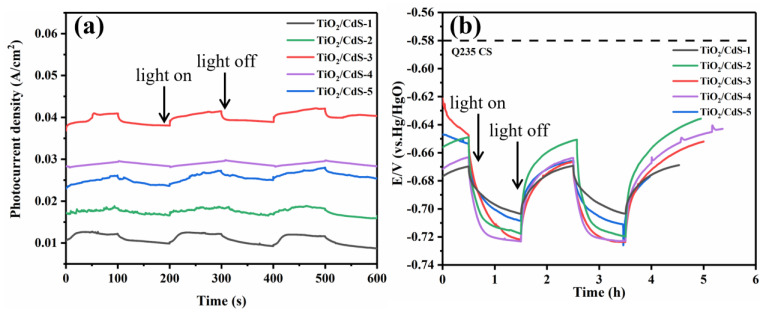
Photocurrent density curve (**a**) and OCP variations curve (**b**) of the Q235 CS coupled with the different composite coating with intermittent visible light being on and off.

**Figure 7 materials-16-03927-f007:**
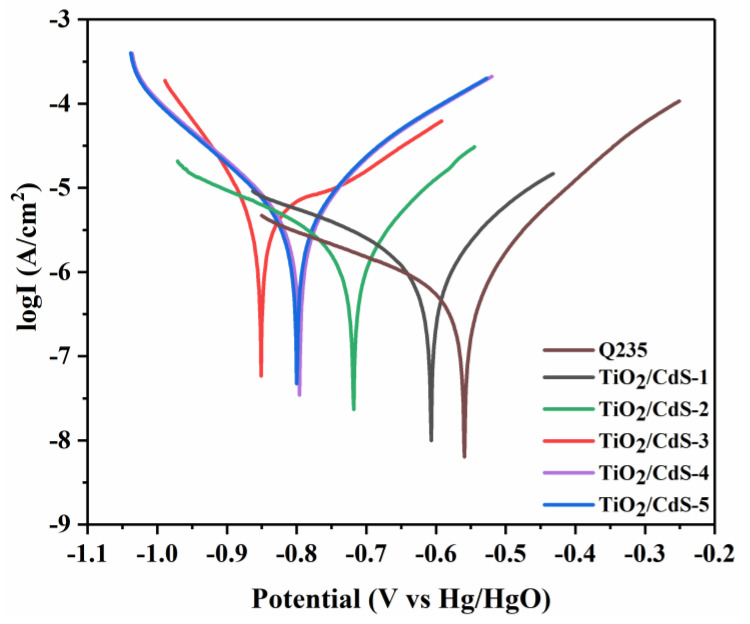
Polarization curve of bare Q235 CS and Q235 CS coupled with different TiO_2_/CdS composite coatings under light illumination conditions.

**Figure 8 materials-16-03927-f008:**
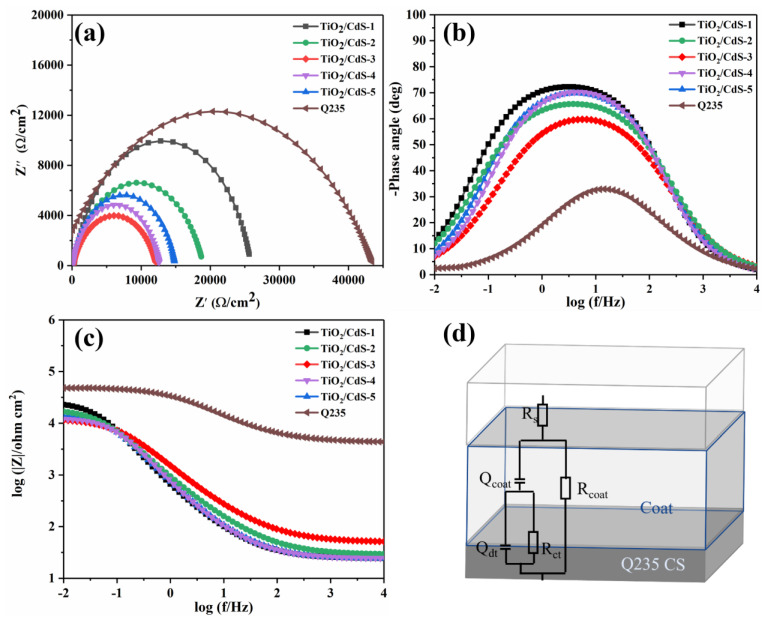
Nyquist plot (**a**) and Bode plots (**b**) of impedance modulus versus frequency of different TiO_2_/CdS samples. (**c**) Phase angle versus frequency of different TiO_2_/CdS samples under light. (**d**) The electrical equivalent circuit models with illumination.

**Figure 9 materials-16-03927-f009:**
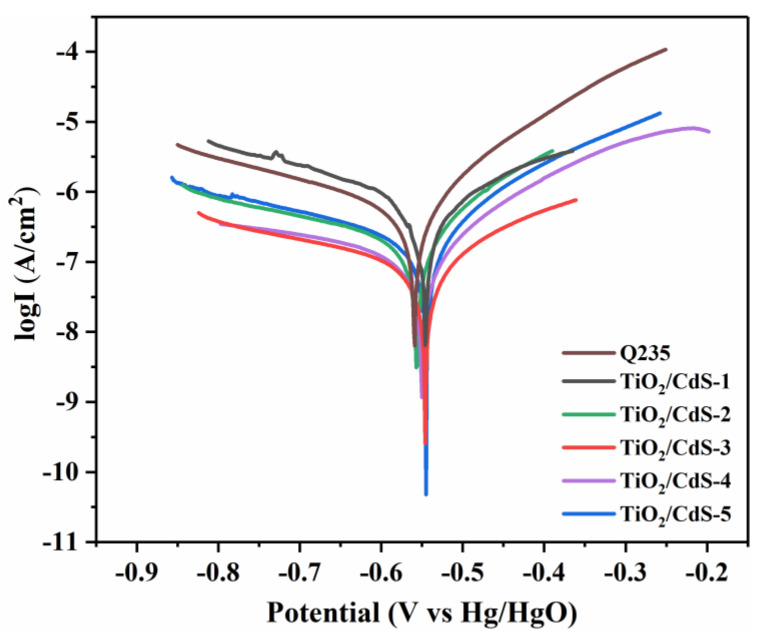
Polarization curve of bare Q235 CS and Q235 CS coupled with different TiO_2_/CdS composite coatings under darkness conditions.

**Figure 10 materials-16-03927-f010:**
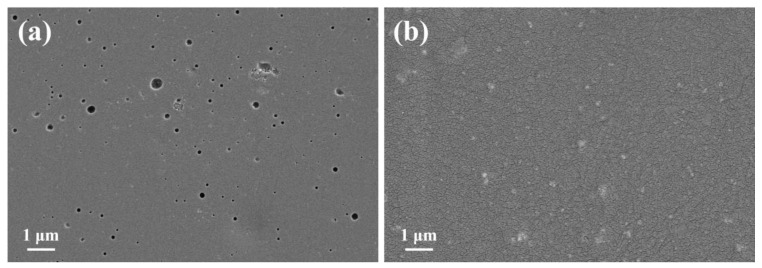
(**a**,**b**) shows the SEM images of neat epoxy and TiO_2_/CdS composite coating.

**Figure 11 materials-16-03927-f011:**
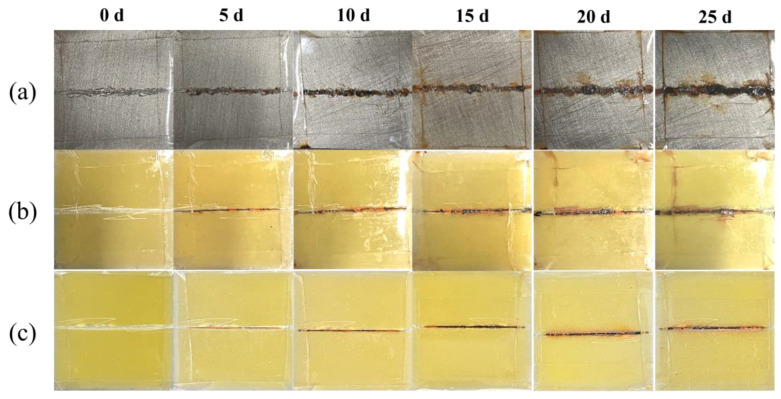
Immersion experiment of Q235 CS coupled with pure epoxy resin coating under dark conditions (**a**) and coupled with TiO_2_/CdS-modified epoxy resin coating under dark conditions (**b**) and light conditions (**c**).

**Figure 12 materials-16-03927-f012:**
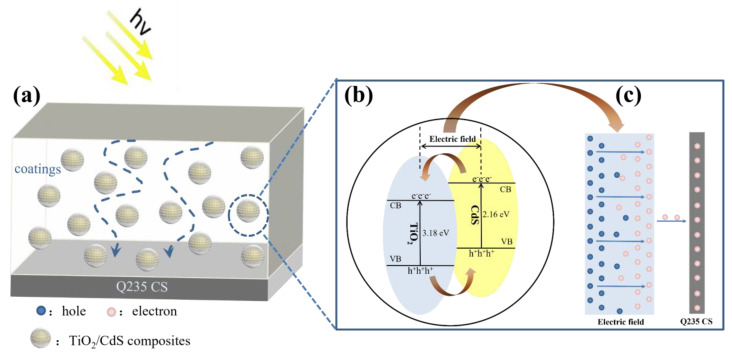
Schematic diagram of photocathodic protection mechanism (**a**–**c**).

**Table 1 materials-16-03927-t001:** The components and performance of different photoelectrodes.

Systems	Metal	Current Density	OCP Drop	Ref.
ZnO	304 SS	26 μA/cm^2^	−0.38 V	[5]
NiSe_2_/TiO_2_	304 SS	283 μA/cm^2^	-	[6]
PbS/TiO_2_	304 SS	3.2 mA/cm^2^	−720 mV	[7]
ZnPc/TiO_2_	304 SS	90 μA/cm^2^	−700 mV	[8]
Ag/SnO_2_/TiO_2_	403 SS	90 μA/cm^2^	−415 mV	[9]
Ag_2_S/ZnS/ZnO	304 SS	19 μA/cm^2^	−0.50 V	[11]
Ag/Ag_3_PO_4_/TiO_2_	304 SS	130 µA/cm^2^	−900 mV	[48]
Zn_3_In_2_S_6_/TiO_2_	Carbon steel	1.91 mA/cm^2^	−0.49 V	[49]
CuInS_2_/TiO_2_	304 SS	118 μA/cm^2^	−0.99 V	[50]
BiVO_4_/CdS	316 SS	304 μA/cm^2^	−330 mV	[51]

**Table 2 materials-16-03927-t002:** Electrochemical impedance parameters of Q235 CS and different TiO_2_/CdS composites in 0.1 M Na_2_S + 0.2 M NaOH electrolyte.

Sample	R_ct_ (ohm·cm^2^)	R_s_ (ohm·cm^2^)	C_dl_ (F/cm^2^)	n	ECC. Models
Q235	45,626	22.73	4.3992 × 10^−5^	0.41	RQR
TiO_2_/CdS−1	25,632	24.18	1.0372 × 10^−4^	0.91	R(Q(R(QR)))
TiO_2_/CdS−2	18,956	28.87	1.1107 × 10^−4^	0.86	R(Q(R(QR)))
TiO_2_/CdS−3	12,216	25.27	8.9728 × 10^−5^	0.80	R(Q(R(QR)))
TiO_2_/CdS−4	12,625	24.30	1.0882 × 10^−4^	0.92	R(Q(R(QR)))
TiO_2_/CdS−5	14,899	23.93	1.1273 × 10^−4^	0.92	R(Q(R(QR)))

**Table 3 materials-16-03927-t003:** Comparison of electrochemical corrosion test parameters between Q235 CS and TiO_2_/CdS anti-corrosive coatings.

Sample	Ecorr (V)	Icorr (A/cm2)	Rp (ohms/cm2)	CR (mm/Year)	PEF%
Q235	−0.559	5.598 × 10^−7^	58,797.6	1.3 × 10^−2^	/
TiO_2_/CdS-1	−0.557	2.455 × 10^−7^	155,261.9	5.703 × 10^−3^	62.13%
TiO_2_/CdS-2	−0.550	2.089 × 10^−7^	176,507.0	4.853 × 10^−3^	66.69%
TiO_2_/CdS-3	−0.545	1.149 × 10^−7^	405,424.3	2.669 × 10^−3^	85.51%
TiO_2_/CdS-4	−0.546	1.298 × 10^−7^	296,483.8	3.015 × 10^−3^	80.17%
TiO_2_/CdS-5	−0.550	1.306 × 10^−7^	294,487.2	3.034 × 10^−3^	80.03%

## Data Availability

Data will be made available upon request.

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
