# Peer review of "Core–Shell Spheroid Structure TiO2/CdS Composites with Enhanced Photocathodic Protection Performance"

_materials, 2023, doi:10.3390/ma16113927_

Round 1

Reviewer 1 Report

Reviewer’s comments

Manuscript Number: materials-2360517

 Title: A core-shell spheroid structure TiO2/CdS nanocomposite with enhanced photocathodic protection performance

 Journal: Materials

Here are some comments: 

1.     It is important to identify all peaks of the EDX.

2.     For each sample (1:1, 1:2, …, 1:5), provide the characterization data (XRD, TEM).

3.     Confirm the nominal ratios (1:1, 1:2, …, 1:5) through experimental testing.

4.     Fit the EIS data (Fig. 8) with an equivalent circuit and plot the fitted data together with the experimental data.

5.     The core-shell morphology is not clear from the TEM images. HRTEM imaging is required to show the crystal planes of both components.

6.     Discuss the correlation between the structural/morphological findings and the electrochemical performance and compare it with other related materials. Literature can be consulted to support this analysis, such as: https://doi.org/10.1016/j.apsusc.2017.05.093 and https://doi.org/10.1016/j.porgcoat.2019.05.011.

7.     Highlight the novelty and improvement of this work compared to previous reports, especially considering that the material has been prepared previously (https://pubs.acs.org/doi/10.1021/am4043068 and https://doi.org/10.1016/j.vacuum.2018.10.082).

Minor editing of English language required

Reviewer 2 Report

The article has reported the synthesis of TiO2/CdS nanocomposite with enhanced photocathodic protection performance. The article has addressed the long-standing issue of corrosion by employing a different approach. The article is suitable for publication in Materials. However, a few small adjustments are needed to make the following important points clear before the manuscript may be published.

1.     It is important to pay close attention to the grammatical errors in the introduction section. The introduction needs to be revised by the authors.

2.     Some of the terms in the caption for Figure 1 began with an uppercase letter. Except for the first word, all letters must be lowercase:

3.     The microstructures of CdS and TiO2/CdS are reported in section 3.1. However, there is a lack of information regarding the microstructure of TiO2. Along with Figure 3, the authors must provide sufficient information the text.

4.     On page 4, lines no 149-150, the authors claimed that many spherical and irregular TiO2 nanoparticles were loaded on the surface of CdS particles. How did the authors arrive at this conclusion? Maybe the CdS nanoparticles are loaded on TiO2.

5.     The authors need to justify whether the particles were nano-sized or micro-sized. The authors have mentioned microstructure characterization in section 3.1. However, they mentioned nanostructure in the text on various occasions. It is very important to determine the particle size to confirm that particles are nano or micro-sized.

6.     In line number 155-156 on page 4, the authors have indicated that the prepared TiO2/CdS composite material was a core-shell spheroid structure, the core was CdS and the shell was TiO2. How the authors confirmed that the core was CdS and the shell was TiO2. It may be the other way around. 

7.     Have the authors mapped Ti from the image shown in Figure 3(c)? If yes, the Ti mapping does not match the image displayed in Figure 3(c). The authors need to explain.

8.     The authors have presented the findings regarding the impact of humidity on HCHO degradation in the results in the discussion section on page 5, line no. 162-174. However, there is no method mentioned for adjusting the humidity inside the chamber. It has to be mentioned in the section on methods.

9.     Line no 208, it should be Figure 4(d) because Figure 5 does not have components. Please recheck.

10.  Because of the size of the Table, Table 3 may be removed and the results to be mentioned in the text.

  The English language needs to be improved.

Reviewer 3 Report

The work presented for review contains interesting research results using a variety of research methods. The research methods were chosen correctly and the results obtained in them correctly interpreted. My only objection concerns the method of determining the energy of the band gap. Please refer, for example, to publication 10.1021/acs.jpclett.8b02892 to determine it correctly. In addition, in almost all materials, two band gap energies can be determined - one for TiO2 and one for CdS. A correction is necessary.

After its implementation, the work can be recommended for publication.

Round 2

Reviewer 1 Report

Reviewer’s comments

Manuscript Number: materials-2360517

 Title: A core-shell spheroid structure TiO2/CdS nanocomposite with enhanced photocathodic protection performance

Journal: materials

The authors have addressed most of the comments in the revised version. The current version could be accepted for publication.

Reviewer’s comments

Manuscript Number: materials-2360517

 Title: A core-shell spheroid structure TiO2/CdS nanocomposite with enhanced photocathodic protection performance

Journal: materials

The authors have addressed most of the comments in the revised version. The current version could be accepted for publication.

Author Response

Thank you for your reviews.

Reviewer 3 Report

Figure 4d still needs correction. Please, check Fig. S3a in https://www.rsc.org/suppdata/jm/c2/c2jm31254a/c2jm31254a.pdf and applied the method in your work. Also another band gap energy arriesing from CdS shoud be designated between 2 and 2.5 eV. Without this, I cannot recommend the work for publication.
